# Partitioning of Rainfall and Sprinkler-Irrigation by Crop Canopies: A Global Review and Evaluation of Available Research

**Meimei Lin** [1] , **Seyed Mohammad Moein Sadeghi** [2] **and John T. Van Stan II** [1,3,*]

[1]  Department of Geology & Geography, Georgia Southern University, Savannah, GA 31419, USA; meimeilin@georgiasouthern.edu

[2]  Department of Forestry and Forest Economics, University of Tehran, Tehran P9CQ+W4, Iran; moeinecohydrologist@gmail.com

[3]  Applied Coastal Research Laboratory, Georgia Southern University, Savannah, GA 31411, USA

*  Correspondence: jvanstan@georgiasouthern.edu

**Abstract:** The role of crop canopies in the global water cycle is a topic of increasing international interest. How much rain and sprinkler-irrigation water are returned to the atmosphere or reach the soils beneath crop canopies, and the pathways of those water inputs at the soil, are linked to agricultural productivity and sustainability. This concise-format review synthesized and evaluated the available, limited, observational data (138 studies) on cropland throughfall, stemflow, and/or interception for >60 crop species covering all major climate types to obtain a global analysis of rainfall and sprinkler-irrigation partitioning by crop canopies. Partitions normalized per unit rain/sprinkler-irrigation (relative fractions, %) vary greatly across crop types with the interquartile range of throughfall, stemflow, and interception being 58–83%, 2–26%, and 11–32%, respectively. Stemflow data distribution across crop types is more often different than for throughfall and interception, contributing to overall variations in the partitioning of rain and irrigation observed to date. Partitions per storm also differ depending on the magnitude of rain or sprinkler-irrigation events and the stage of crop growth. Furthermore, throughfall and stemflow input patterns at the soil surface and subsurface may erode soils through different physical processes (i.e., throughfall droplet impact/splash versus scouring by stemflow); however, more research is needed to elucidate the underlying mechanisms and overall impacts. Finally, comparative analyses of partitions among croplands, shrublands, and forests indicate that crop canopies partition rain inputs differently and that there is a lack of studies in croplands. Hence, we suggest that future effort should be directed to the partitioning of rainfall and sprinkler-irrigation by canopies in agricultural settings.

**Keywords:** throughfall; stemflow; interception; agriculture; meta-analysis

## 1. Introduction

Global crop production and its connection to water resources are foundational to human well-being. Both crop yields and hydrological processes depend, in large part, on climate variables; yet, occurrences of climate extremes have increased over the past century and are projected to become more frequent and severe for many regions [1,2]. Rainfed croplands may be impacted by changes in storm conditions due to hydrologic intensification, or acceleration of the water cycle [3,4]. When precipitation is scarce, or intense but infrequent, irrigation may be applied to croplands; however, irrigation water supplies also have environmental limitations that are influenced by climate change [5,6], amongst other factors. In many croplands, sprinkler systems are commonly used to apply irrigation above, near, or within crop canopies [7]. Thus, a precise understanding of the

amount of rainfall or sprinkler-irrigation returned to the atmosphere by interception or reaching cropland soils, and their pathways at the surface and through the subsurface are arguably valuable to sustainable agriculture.

Only a portion of the droplets that contact crop canopies reaches the surface, as some water is evaporated during passage through the canopies (interception). Droplets that are not lost during this interception process reach soils as a drip flux (throughfall) or as a flux of water down the surface of stems (stemflow). Extensive research literature examines evaporation from croplands [8], but comparatively, little research has been conducted in croplands quantifying the portion of net rainfall/irrigation reaching soils as throughfall versus stemflow—especially compared to throughfall and stemflow research in forests [9]. Indeed, over the past 150 years, modeling frameworks for the partitioning of rainfall into interception, throughfall, and stemflow have been primarily based on observational studies in forests and shrublands [9–12]. This is problematic as plant canopy structure in agricultural settings differs markedly from forests and shrublands, especially for herbaceous crops; therefore, croplands are expected to partition rainfall and sprinkler-irrigation waters differently.

As a result, a review and evaluation of observational studies describing crop canopies' interception, throughfall and stemflow under rainfall and sprinkler-irrigation conditions is merited. This article is a condensed review of 138 studies reporting these observations to date, to (i) discuss the relevance of these synthesis and comparative results to surface (i.e., soil erosion) and subsurface processes (i.e., soil moisture) in croplands, (ii) compare these observations to several hundred forest and shrubland studies recently synthesized [9], and (iii) identify current knowledge gaps and suggest future directions for rainfall and irrigation partitioning research in cropland ecosystems.

## 2. Literature Search and Data Compilation Methods

This meta-analysis relies on data compiled from a synthesis of published studies that reported interception, throughfall, and stemflow from crop canopies. Several databases to which the Georgia Southern University library is subscribed were searched (Web of Science, BIOSIS, Current Contents Connect, and The Scientific Electronic Library Online) without a date restriction (i.e., 1864–present) for "crop" AND "stemflow" OR "throughfall" OR "interception". Studies were excluded if they were not scholarly peer-reviewed publications or if their abstracts did not report a result on throughfall, stemflow, or interception. Few studies matched these criteria in the subscription databases (i.e., this is the database from Sadeghi et al. [9] which only contains 22 cropland studies). Thus, to expand the meta-analysis, we performed a targeted search for studies reporting results on interception, throughfall, and stemflow for specific crop species. As the definition of which plants are "crops" can vary, we relied on the list of crops within the United Nations Food and Agricultural Organization (UN-FAO) crop product types [13]. We then searched for additional studies focused on UN-FAO crop species that were missing from the subscription databases via Google Scholar. The Google Scholar search was performed using "Latin name" OR "common name" AND "stemflow" OR "throughfall" OR "interception" without date restrictions. Sometimes Google Scholar returned a substantial amount of search results that appeared mostly tangential; so, when Google Scholar returned >10 pages of results, only the first 10 pages were checked to ensure that results were no longer on the search topic. Checking a result, specifically, entailed clicking the link and querying the abstract (when available) for the key terms (throughfall, stemflow, and interception). Non-English publications were located based on the authors' previous knowledge or personal communications with international collaborators, translated and incorporated into the database. This search resulted in 138 papers.

Digitization of the throughfall, stemflow, and interception data was performed using Tabula 1.2.1 (https://tabula.technology/), when possible, and manually when the automated digitization routine failed. To ensure that the Tabula software correctly digitized the datasets, observations in each table were checked manually (all data were able to be manually reviewed by at least one of the authors). Most Tabula errors encountered were formatting errors. Canopy precipitation partitioning research, in general, has no consensus on standard units for reporting throughfall, stemflow, and interception.

Thus, to enable study intercomparison, all data were normalized as per unit rain or irrigation across the crop canopy (%, relative fractions of above-canopy water). The final synthesis is provided in Table S1 of the supplemental materials.

Data were analyzed to compare precipitation partitions across UN-FAO crop types [13] and for comparison of crops against more commonly studied plant types in the precipitation partitioning literature, shrubs and trees [9]. Due to the limited data for individual species, comparative analyses of rainfall and sprinkler-irrigation partitioning were performed based on the UN-FAO crop product types [13]. Descriptive statistics and violin plots were extracted from the dataset for each UN-FAO crop type and type of plant. Since sample size (i.e., synthesized observations) of rainfall or irrigation partitions per UN-FAO crop type and type of plant differed, *k*-sample Kolmogorov–Smirnov tests were used to test for differences in the distribution of interception, throughfall, and stemflow. Tests were only performed when observation sample sizes were large enough per group to develop a probability density function (these instances are indicated in figures when present). Kruskal–Wallis tests were also performed. All statistical analyses were performed in RStudio 1.2.5 (PBC, Boston, MA, USA). Figures from selected papers were digitized to scale using the single-scan auto-trace routine in Inkscape 1.0 (http://www.inkscape.org/); then, traces were manually inspected/corrected if necessary.

## 3. Rainfall and Irrigation Partitioning by Crop Canopies

The synthesis dataset contains >60 crop species (Figure 1; Table S1 of supplemental materials). Studies reported observations from three experimental conditions: sprinkler-irrigation (11 studies), simulated rainfall (21 studies), and natural rainfall (111 studies)—note that this number exceeds the 138 studies reviewed because some studies tested multiple experimental conditions. Only one of the UN-FAO crop types was tested under all experimental conditions, cereals. All other UN-FAO crop types were either tested exclusively under a single experimental condition or, when tested under two experimental conditions, there were no significant differences detected between the median or distribution of reported partitioning percentages ($p > 0.05$ for Kruskal–Wallis and Kolmogorov–Smirnov tests). For the above reasons, the following analyses include all experimental conditions together; however, we acknowledge that, considering that only a few studies have directly compared these experimental conditions [14–17], there may be differences in how crop canopies partition natural rainfall and sprinkler-irrigation not yet exposed by the extant literature.

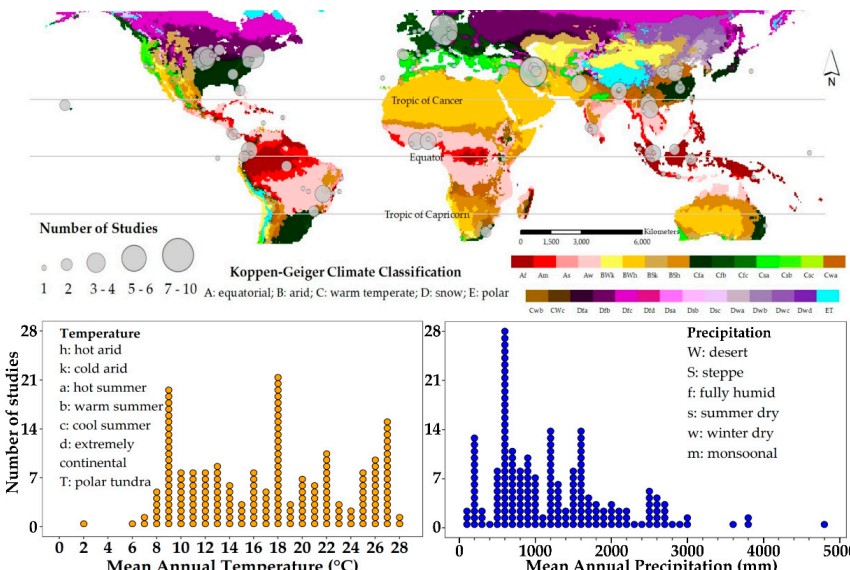

**Figure 1.** Map of locations and climate conditions for all study sites included in this meta-analysis of observations or estimates of throughfall, stemflow, and interception for croplands.

Observations of rainfall and irrigation partitioning by crop canopies have been reported across all major climate types where crops are typically cultivated (Figure 1). The range of mean annual temperature and precipitation across sites was 2.2–28.4 °C and 95–4750 mm year$^{-1}$, respectively. Few sites identified in this literature synthesis had mean annual temperatures <9 °C or mean annual precipitation >3000 mm year$^{-1}$ (Figure 1). Studies from very wet sites, >3000 mm year$^{-1}$, were dominated by tall vegetation in agroforestry settings, such as plantations of cashew [18], guava [19], banana [20], and cacao [21]. Cold climate studies primarily focused on shorter cereal and forage rotation crops, such as maize [22–24], barley [25], potatoes [26,27], and alfalfa [28,29]. Two common crops have been examined for rainfall and irrigation partitioning across several climates—wheat [25,30–33] and maize [15,34–37]. Coffee plantations have received comparatively significant research attention on this topic within the tropics [38–42].

Few observations were found in past literature for three of the nine crop types: leguminous crops, root/tuber crops, and vegetables/melons (Figure 2). The authors were only able to find one study reporting all rainfall partitions for any crop classified as "vegetables or melons" [43]. Rainfall partitioning research on root/tuber crops has focused on potatoes [25–27,36] and cassava [44]. Similarly, for leguminous crops, only two species have been researched on this topic: alfalfa and clover [28,29,45]. Across all studies, the interquartile range of throughfall, stemflow, and interception is expansive—58–83%, 2–26%, and 11–32% of rainfall and sprinkler-irrigation, respectively (Figure 2)—demonstrating how variable these partitions can be across crop types. Data distributions are more often different between stemflow across crop types than for throughfall or interception (Kolmogorov–Smirnov test results provided in Table S2 of the Supplemental Materials). Significant differences in stemflow across crop types appear to be the largest driver of changes in overall partitioning (Figure 2). The greatest stemflow producers were the cereal crops (median = 33.4%), and the lowest stemflow producers were crop types dominated by trees, such as fruits, nuts, cloves, and coffee (group medians ranging from ~2 to 5% of rainfall).

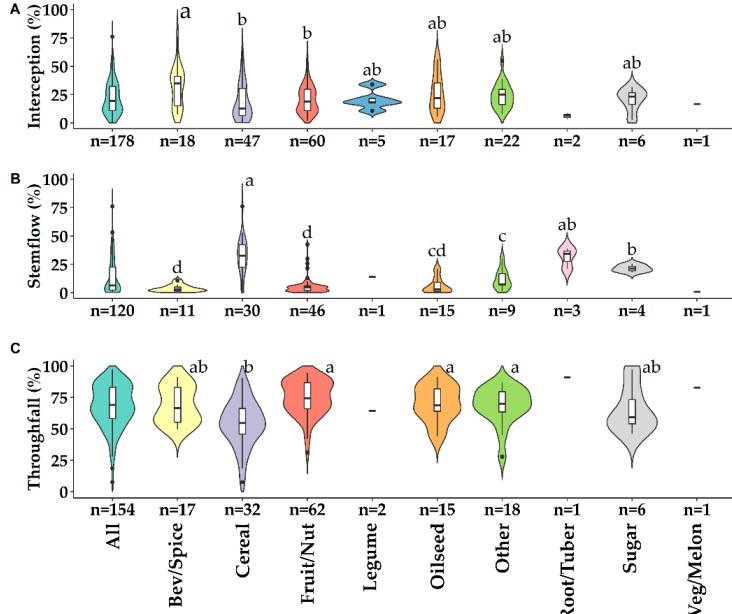

**Figure 2.** Violin plots of observed relative (**A**) interception, (**B**) stemflow, and (**C**) throughfall normalized per amount of rainfall or sprinkler-irrigation and grouped by the UN-FAO crop product types. Symbols indicate the median (horizontal line), interquartile range (box), non-outlier range (vertical line), outliers (dots), and probability density (violin). Where sample sizes were large enough per group to develop a probability density function, Kolmogorov–Smirnov tests assessed differences in probability distributions (Table S2 in Supplemental Materials) and alphabetical superscripts indicate significant differences ($p < 0.05$).

## 4. Storms and Growth Stage: Temporal Variability of Rain and Irrigation Partitioning

Crop canopies are reported to have maximum water storage capacities ranging from 0.6 to 6.0 mm event$^{-1}$, and may store up to 11.2 mm event$^{-1}$ for agroforestry settings with multiple canopy levels [46]. The amount of this maximum water storage capacity available for any crop canopy immediately before a rain or irrigation event depends on the degree to which the canopy has dried during the interevent dry period [47]. As a result, substantial interevent variability occurs for canopy partitioning. This variability in the fraction of throughfall or stemflow ($y$, %) and interception ($y_I$, %) has been well described across vegetation types by the exponential formulae:

$$y = \alpha \, (1 - e^{-\beta P}) \tag{1}$$

$$y_I = \alpha \, e^{-\beta P} + c \tag{2}$$

where $P$ is event magnitude (mm event$^{-1}$) and, for throughfall or stemflow fraction, the fitting coefficients $\alpha$ and $\beta$ are the maximum throughfall or stemflow fraction (%) and the rate at which this maximum fraction is reached with increasing event size, respectively (Figure 3). For the interception fraction, $\alpha$ and $\beta$ are the maximum interception fraction (%) and the rate at which the minimum interception fraction ($c$) is reached with increasing event size, respectively (Figure 3). These exponential fits explained significant interstorm variability across crop types, from potatoes [26,27] and maize [48] to trees [49].

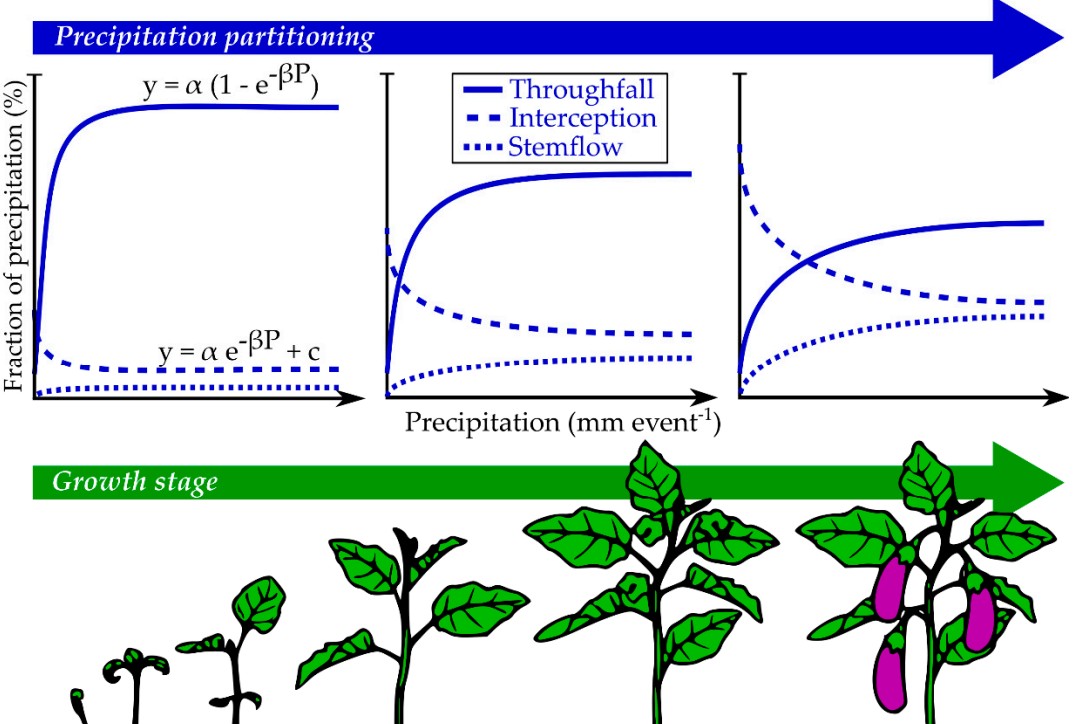

**Figure 3.** Conceptual model showing the interevent variability of precipitation partitions (throughfall, stemflow, and interception) across crop growth stages—e.g., [15,48] and references discussed in the text. Exponential formulae often fit to these trends [50,51] are provided in the upper left panel.

Fitting coefficients ($\alpha$, $\beta$, and $c$) vary across growth stages as a function of increased maximum water storage capacity [50] and changes in the inclination angle of canopy elements, particularly leaves for herbaceous crops [52–54]; however, the orientation of fruit, tassels, etc., and branches (for tree orchards) can alter water drainage patterns through canopies [48,55,56]. As maximum water storage

capacity is a function of the canopy area available to capture and hold rain or sprinkler-irrigation waters, the magnitude of the maximum interception fraction (during storms too small to fully saturate the canopy) can depend on the growth stage (Figure 3). For example, a mature 1.2 m *Melilotus officinalis* (sweet clover) was observed to intercept 92–94% of small rainstorms [57]. Variability in throughfall fraction with event size generally responds oppositely to the interception fraction but, importantly, throughfall fractions may diminish across growth stages more than would be expected from changes in interception alone—due to redirection of canopy waters to stemflow (Figure 3). Increased canopy area can coincide with (i) increased leaf or branch inclination angle [52–54] and/or (ii) increased leaf hydrophobicity as cuticular waxes establish, which may drain greater water fractions toward the stem and increase the stemflow fraction as crop growth progresses [14,58,59]. Interestingly, we note that the maximum stemflow fractions per storm in mature croplands are often of similar magnitude to interception fractions (Figure 3, top right) and can exceed throughfall fractions (e.g., see review of precipitation partitioning by maize canopy in [15]). Several studies on erectophile, mature crop canopies have documented ≥50% of rain and irrigation reaching the soil surface as stemflow [27,37,58,60–62].

## 5. Surface and Subsurface Relevance of Throughfall and Stemflow in Croplands

The partitioning of rainfall and irrigation by crop canopies results in complex fine-scale throughfall and stemflow input patterns at the surface that can influence surface and subsurface hydrologic and biogeochemical processes [20,37,63,64]. Starting from the top of the soil, throughfall and stemflow can erode soils, but through different physical processes. Splash forces from throughfall droplets can dislodge soil particles. A portion of throughfall consists of large water droplets (3–6 mm in diameter—e.g., [65]) formed from water accumulation on leaves and released at regular intervals during steady irrigation or rainfall. These throughfall droplets can be many times larger than typical rain droplets (1–3 mm: [66]) and, thereby, much heavier (i.e., a spherical droplet's mass is proportional to the diameter cubed). This results in a greater erosive potential for impacts from throughfall drip points than from rain (or light sprinkler-irrigation) droplets. The erosive potential of throughfall drip points is enhanced by their temporal persistence [67], as the impact location of rain and light sprinkler-irrigation droplets temporally varies, yet canopy areas draining to a throughfall drip point route water to nearly the same location throughout a storm [66]. For these reasons, splash erosion from throughfall beneath vegetation has been observed to be 3–9 times greater than for open rainfall for a rubber plantation, depending on storm magnitude [68].

Stemflow, being a rivulet flow that is transferred from a near-vertical stem surface to a near-horizontal soil surface, can dislodge soil particles via scouring. Stemflow-induced runoff has been reported for maize and sorghum under intense simulated rainfall [69]. Of note, maize has also been observed to drain large fractions (78%) of natural net rainfall as stemflow [70], which can result in the concentration of rainwater from a ~0.5 m$^2$ plant$^{-1}$ canopy area by >500 times to ~0.0005 m$^2$ plant$^{-1}$ of soil area immediately surrounding the stem base. Recently, Zhao et al. [71] observed that stemflow contributed to rill erosion around the base of small-diameter stems under controlled simulations. The concentration of stemflow to near-stems soils depends on the canopy area; thus, the large canopies of fruit and nut trees may be able to generate substantial stemflow volumes from a small fraction of rainfall. For example, stemflow from ~7% of rainfall (on average) in a macadamia orchard eroded 3.8 tons ha$^{-1}$ y$^{-1}$ of soil, lowering the soil surface around stems by 6.5 mm m$^{-2}$ y$^{-1}$ [59]. Stemflow-related runoff from cereals has been considered a "negligible" contributor to soil erosion compared to throughfall splash erosion in the past [69]; however, no studies have considered the role that stemflow-induced runoff may play in enhancing splash erosion by throughfall and rain/sprinkler-irrigation droplets. Soil particles ejected by splash impacts may be carried off-site by stemflow-related runoff pathways. Additionally, in the presence of shallow flowing water, splash impacts from throughfall (discussed earlier) can create rapid lateral jets [72] that may dislodge soil particles that would not have been dislodged by runoff flows alone (Figure 4).

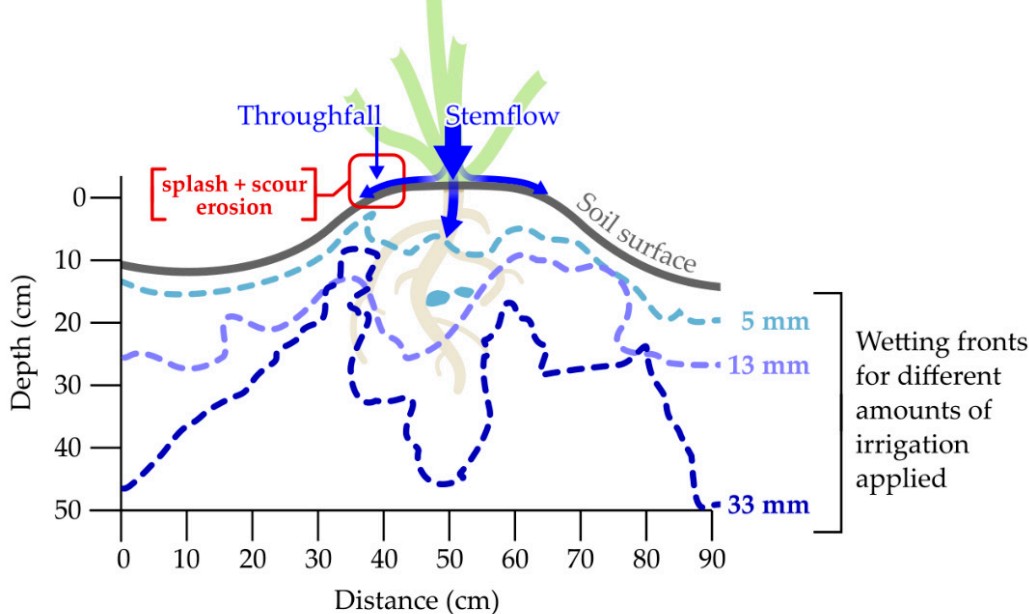

**Figure 4.** Digitized wetting fronts in a Plainfield loamy sand soil (Typic Udipsamment; sandy, mixed, mesic) observed beneath a potato crop (*Solarium tuberosum* L, var. Russet Burbank) with increasing sprinkler-irrigation amounts (adapted from [26]). A preferential infiltration feature visibly forms with increasing irrigation beneath the stem base, likely due to stemflow infiltration along roots. Note that a portion of stemflow can become surface run-off and potentially enhance throughfall droplet-related erosional processes (through creation of rapid lateral jets from splash-on-film [72] or by transporting impact-related particles [59]).

The amount of runoff and erosion from throughfall and stemflow in agricultural settings depends on soil properties and topography [64,66,69]. For example, soil physicochemical variables control the porosity, thickness, and water repellency of "surface seals" that may result from the impacts of throughfall, rain, and sprinkler-irrigation droplets [73–75]. Given the impact energy associated with throughfall drip points (discussed earlier), it is not surprising that reduced infiltration due to throughfall-induced soil surface seals has been observed in croplands, including temporary ponding in the furrows of ridge-and-furrow settings (e.g., [76–78]). Alternatively, soil conditions may also allow preferential infiltration of stemflow, as has been reported for both short crops [17,26,79] and tall crops, such as an olive grove [80]. The most extensive study of net precipitation was performed for potatoes where a hydrologic tracer dye (Rhodamine WT) was applied and mapped with increasing sprinkler-irrigation amounts [26] (results digitized and compiled in Figure 4). The observed dye patterns indicated that sprinkler-irrigation could preferentially infiltrate at the base of plants via stemflow (Figure 4). The depth of the stemflow-induced preferential infiltration feature increased directly with sprinkler-irrigation amounts, reaching 50 cm deep after 33 mm of irrigation (Figure 4).

Although little research has investigated and reported (i) the fraction of throughfall and stemflow that runs off or infiltrates, (ii) where those hydrologic fluxes go, or (iii) developed a mechanistic understanding of this partitioning, the pursuit of these aims may have benefits that extend beyond water management itself—benefitting our understanding of the fate and transport of nutrient or pest control agents in croplands [64].

## 6. Cropland Precipitation Partitioning Significantly Differs from Shrubs and Forests

The fractions of interception, throughfall, and stemflow observed beneath crops show apparent differences compared to observations from shrublands and forests (Figure 5). Considering the frequency distribution of interception estimates, croplands showed a wide distribution that was positively skewed and bimodal. The two modes in cropland interception fraction reflect differences in woody

and herbaceous crops, with the mode of lesser relative frequency (30–35% bin) consisting primarily of cereal and cover crops [29,57]. The greater relative frequency mode for crop interception fraction (10–15% bin) consisted primarily of the more-often researched tree crops such as apple orchards [81], cocoa [82,83], coffee [38], and coconut [84] and is close to the mode for forest interception fraction (15–20% bin) but low compared to the mode for shrublands (Figure 5, top).

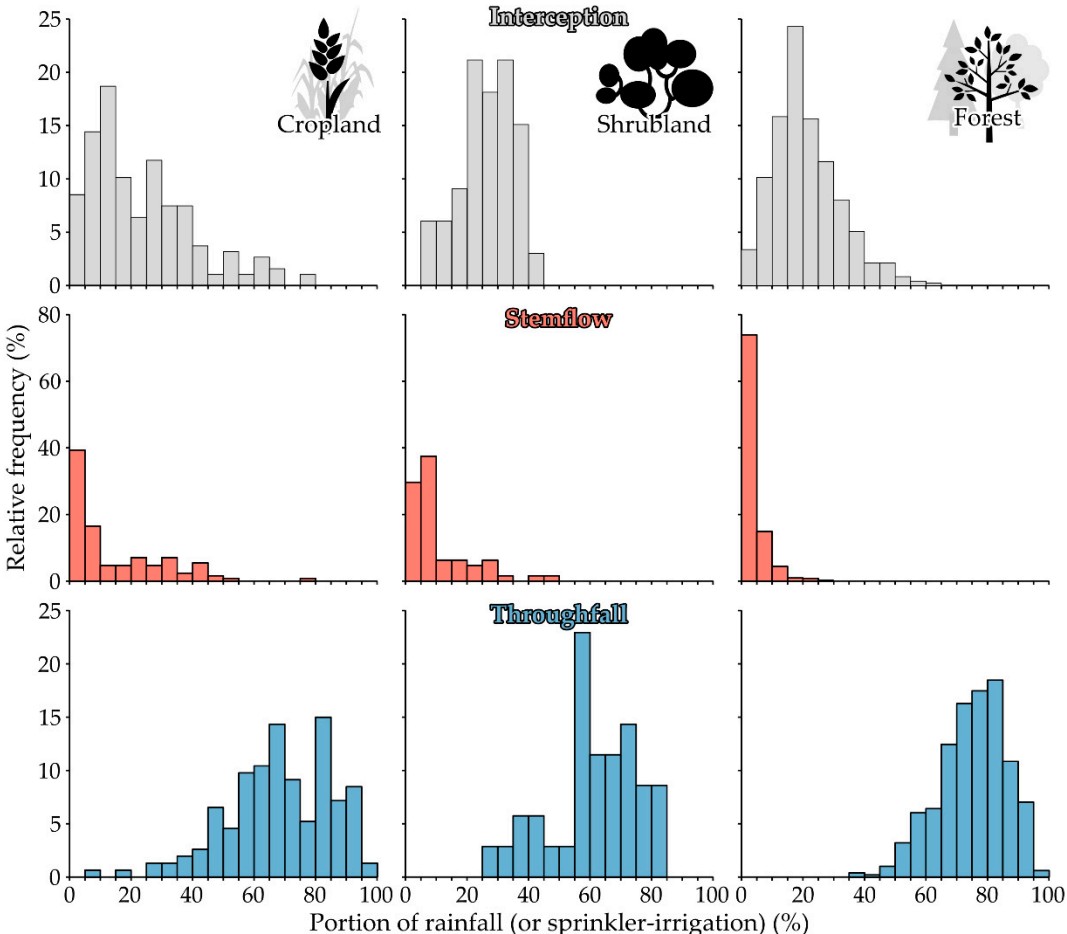

**Figure 5.** Relative frequency histograms of observed relative interception, stemflow, and throughfall for all crops, shrubs, and forests show rainfall partitioning patterns across vegetation types. Shrub and forest data from a recent synthesis of these data [9].

Interception fraction from woody plants in agricultural settings may be lower compared to natural forests (Figure 2b vs. Figure 5, middle) as a result of the greater stemflow generation observed beneath managed tree canopies [56,85]. In fact, histogram distributions of stemflow fractions for all three settings were very positively skewed with drastic differences in the distributions. Crop stemflow data had the widest range (0.2–76%), compared to shrubs and forests (Figure 5, middle), again likely reflecting the diversity of plant morphologies represented in croplands (Figure 2b). The stemflow fractions from cropland studies were generally larger than for shrublands and natural forests (Figure 5, middle). Opposite to the trend observed in the stemflow data, throughfall fractions showed negatively skewed distributions for all settings, with cropland studies reporting lower throughfall fractions than shrublands or forests (Figure 5, bottom). In light of the comparisons in Figure 5, it is clear that crop canopies in agricultural settings can partition rain and sprinkler-irrigation waters differently from the woody settings under which current throughfall, stemflow, and interception theory has been developed and tested.

## 7. Conclusions

Understanding rainfall and sprinkler-irrigation partitioning patterns and variations in croplands may be relevant to global crop productivity and sustainability, as well as water resource management, especially given current global climate change projections. Globally, croplands have had the highest proportion of water input (precipitation plus irrigation) becoming recharge, compared to grasslands, woodlands, and scrublands [86]; hence, study on the ecohydrological processes of cropland ecosystems that control the amount of water inputs to the surface is merited. As evaporation of intercepted water also affects the surface energy budget, work is merited on the role of interception on cooling the atmosphere and crop itself. In this review, we synthesized and evaluated 138 observational studies on crop canopies' interception, throughfall, and stemflow in terms of variability across storms and growth stages, relevance to surface and subsurface hydrologic and biogeochemical processes, and comparisons to forest and shrubland studies. The fractions of throughfall, stemflow, and interception vary greatly across crop types, storm magnitudes, and crop growth stages—this is especially true for stemflow. Fine-scale examination of this partitioning showed that throughfall and stemflow reaching the soil surface can potentially lead to soil erosion through various physical processes; however, there is limited research in this vein. Hence, research is needed to track where throughfall and stemflow go and how they impact runoff and erosion at the soil surface and subsurface. Such information may be valuable for improving understanding of the fate and transport of nutrient or pest control agents in croplands. In addition, most theoretical and statistical modeling of precipitation partitioning has been focused on forests and shrublands due to limited studies in croplands. Therefore, an overarching conclusion of this meta-analysis is that research on the partitioning of rainfall and sprinkler-irrigation by canopies in agricultural settings is needed to both advance theoretical models and develop practical water resource applications.

**Supplementary Materials:** The following are available online at http://www.mdpi.com/2306-5338/7/4/76/s1, Table S1: alphabetized literature values of relative canopy interception, stemflow, and throughfall. Table S2: results of the Kolmogorov–Smirnov tests used to test for differences in the distribution of interception, throughfall, and stemflow between paired crop types and for all crop data combined.

**Author Contributions:** Conceptualization, M.L. and J.T.V.S.II; data curation/synthesis, S.M.M.S.; methodology, M.L. and S.M.M.S.; formal analysis, all authors; writing—original draft preparation, M.L. and J.T.V.S.II; writing—review and editing, all authors.; visualization, M.L. and J.T.V.S.II. All authors have read and agreed to the published version of the manuscript.

**Funding:** This research received no external funding.

**Acknowledgments:** We gratefully acknowledge DAR Gordon for assistance in compiling the forest precipitation partitioning database and we thank the peer reviewers.

**Conflicts of Interest:** The authors declare no conflict of interest.

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
