# Peer review of "Partitioning of Rainfall and Sprinkler-Irrigation by Crop Canopies: A Global Review and Evaluation of Available Research"

_hydrology, doi:10.3390/hydrology7040076_

Round 1

Reviewer 1 Report

Good job! It was informative and thought provoking. A person in my group was preparing to work on this in cotton and sorghum. I hope this is published quickly.

Just one little thing:
Lines 45-46 should be changed to read:  "...a flux of water down the surface of stem 45 (stemflow)."
Yes, this was addressed later (Line 212) but was a bit confusing. Some might confuse stem for sap flow. I did at first.

Otherwise I could find no major problems. I did not go through each and every reference, and the stats were a bit beyond me.

Author Response

Good job! It was informative and thought provoking. A person in my group was preparing to work on this in cotton and sorghum. I hope this is published quickly.

Just one little thing:
Lines 45-46 should be changed to read:  "...a flux of water down the surface of stem 45 (stemflow)."
Yes, this was addressed later (Line 212) but was a bit confusing. Some might confuse stem for sap flow. I did at first.
Otherwise I could find no major problems. I did not go through each and every reference, and the stats were a bit beyond me.

Response: Thanks, we addressed Lines 45-46 to read:  "...a flux of water down the surface of stems (stemflow)."

Author Response

Response: Thank you! Although we were not required to address these reviewer comments, we read through this review and incorporated a number of suggested revisions:

Figure 1 – we labeled the y-axis "Number of studies"

Figure 3 - we cited some example crop studies that show these trends (Nazari et al.; Zheng et al., 2018, etc)

Figure 4 - in caption, we added the soil type

Figure 5 - x-axis, we stated "sprinkler irrigation"

L12: in abstract, we corrected to read "How much rain and irrigation water is returned to the atmosphere or reaches the soils beneath crop canopies..."

L13: in abstract, we deleted the term "directly" as suggested

L17: in abstract, we changed to "per unit rain/sprinkler irrigation...” We also made sure that every mention of irrigation is 'sprinkler'

L41: we changed to "...sprinkler-irrigation returned to the atmosphere by interception or reaching cropland soils..."

L282-283: we changed to read "partitioning patterns and variations in croplands may be relevant to global crop productivity and sustainability..."

L287: we added the sentence as you suggested "...water inputs to the surface is merited. As evaporation of intercepted water also affects the surface energy budget, work is merited on the role of interception on cooling the atmosphere and crop itself. In this..."

Reviewer 3 Report

Research on "interception" in farmland is methodically difficult.
The authors rightly note the complexity of such activities and identify current knowledge gaps and suggest future directions for rainfall and irrigation partitioning research in cropland ecosystem
The article is a description and analysis of 138 rainfall penetration studies. This is a very important work worth publishing in MDPI.

I do not find any methodological or substantive comments.

Author Response

Response: Thank you!

Reviewer 4 Report

            The manuscript entitled “Partitioning of rainfall and sprinkler-irrigation by crop canopies: A global review and evaluation of available research” (reference number hydrology-926044), authored by M. Lin, S.M. Moein Sadeghi and J.T. Van Stann, II presents a meta-analysis of the information currently available on the topic of rainfall and irrigation partitioning in croplands. The authors performed a literature search and analyzed the data from the 138 papers they selected. The study is clearly described and concisely reported. The manuscript is interesting and fits within the scope of the Hydrology journal.

            The introduction section provides a brief and justified overview of the problem and why this study is needed. The methodology employed is correct and well described. The results are clearly presented and discussed, while the conclusions are in agreement with the objectives proposed.

            English is correct, although I detected some unclear sentences that need clarification.

In view of this, I recommend a minor revision of this manuscript prior to its acceptance for being published in Hydrology.

Specific comments to authors:

Abstract:

The abstract is concise and explains clearly the objectives and the main results from the review work undertaken. However, I have three concerns:

Line 13: “the patterns of those water inputs at the soil”, this is unclear. What do you mean by “patterns”? Do you mean infiltration? Do you mean runoff? Do you mean spatial or temporal patterns? I would use the word “pathways”, but I am not sure that this is what you meant. Please, clarify this point.

Line 22: Include “the” before “stage”.

Lines 22-24: This sentence is confusing. Please, re-phrase it.

Keywords:

“precipitation partitioning” can be removed as, in the title, you already stated “Partitioning of rainfall”.

Introduction:

This section explains well the reasons for performing this review and the objectives that the manuscript tries to reach.

Line 37: What do you mean by “hydrologic intensification”? Please, clarify.

Line 39: Not only by climate change, but also by restrictions imposed by water users associations. I would include “amongst other factors” at the end of this sentence.

Line 42: “and their surface patterns”, this is unclear from my point of view. What do you mean by “patterns”? Do you mean infiltration? Do you mean runoff? Do you mean spatial or temporal patterns? I would use the word “pathways”, but I am not sure that this is what you meant. Please, clarify this point.

Literature Search and Data Compilation Methods:

In this section, you could indicate the final number of papers that you used in the meta-analysis. For instance, at the end of the section, say something such as “This search produced 138 papers”.

Line 85: “the Tabula software correctly digitized the datasets” instead of “the Tabula software did not incorrectly digitized the datasets”.

Line 86: What do you mean by “spot-checks”?

Lines 88-89: This sentence (“Since canopy…”) seems to be incomplete. Please, check.

Line 91: Why mentioning Table S2 prior to Table S1?

Line 103: “selected” instead of “select”.

Results and Discussion:

Line 115: The Kruskal-Wallis test must have been mentioned in the previous section, when describing the analysis of data that you performed.

Figure 1: In the legend of the bottom right panel, use “Precipitation” instead of “Precipiattion”.

Line 143: I suggest swapping the order of the supplementary tables.

Line 184: “reduce small rainstorms”, what do you mean? The intensity of the rainfall? The speed in which raindrops reach the soil surface? The volume of rainfall reaching the soil surface? Please, clarify.

Line 186: “stages” instead of “stage”.

Line 189: Include “fractions” after “greater water”.

Line 190: “crop growth” or “crop development” instead of “growth stage”.

Lines 221-223: Please, check this sentence. It seems to be incorrect. Maybe, remove “Observations of”.

Line 230: “surface seals”, do you mean “surface crust”?

Line 266: I suggest removing “show clear differences in rainfall partitioning across vegetation types”.

Line 280: If “theory” is the subject, then it should be “has been developed” instead of “have been developed”.

Conclusions:

Line 288: I suggest removing “(all that we were able to find)”.

References:

Lines 317-318: This reference is incomplete. Volume and page numbers are missing.

Lines 322-324: This reference is incomplete. Volume and page numbers are missing.

Lines 335-337: This reference is incomplete. Page numbers are missing. If it is a book, publisher and location of the publisher are also missing.

Lines 338-339: This reference is incomplete. Page numbers are missing. If it is a book, publisher and location of the publisher are also missing.

Line 354: Page number is missing.

Line 357: “Theobroma cacao” should be in italics.

Lines 365-366: This reference is incomplete. Page numbers are missing. If it is a book, publisher and location of the publisher are also missing.

Line 391: “Coffea arabica” should be in italics.

Line 402: “Coffea arabica” should be in italics.

Lines 409-410: Please, do not use capital letters for the surnames of the authors. Include number of pages.

Line 431: “Punica granatum” should be in italics.

Line 484: “Theobroma cocoa” should be in italics.

Line 488: “Cocos nucifera” should be in italics.

Line 492-493: Please, check this reference. The abbreviated title of the journal and the number of pages seem to be incorrect.

Author Response

Response: Thank you! We have addressed all your comments accordingly as hown below:

Line 13: “the patterns of those water inputs at the soil”, this is unclear. What do you mean by “patterns”? Do you mean infiltration? Do you mean runoff? Do you mean spatial or temporal patterns? I would use the word “pathways”, but I am not sure that this is what you meant. Please, clarify this point.

Response: Yes, we meant pathways. We changed the sentence to "How much rain and sprinkler-irrigation water is returned to the atmosphere or reaches the soils beneath crop canopies, and the pathways of those water inputs at the soil, are linked to agricultural productivity and sustainability."

Line 22: Include “the” before “stage”. 

Response: Corrected.

Lines 22-24: This sentence is confusing. Please, re-phrase it.
Response: Agree. We rephrased it to be "Partitions per storm also differ depending on the magnitude of rain or sprinkler-irrigation events and the stage of crop growth. Furthermore, throughfall and stemflow input patterns at the soil surface and subsurface may erode soils through different physical processes (i.e., throughfall droplet impact/splash versus scouring by stemflow); however, more research is needed to elucidate the underlying mechanisms and overall impacts."

Keywords: “precipitation partitioning” can be removed as, in the title, you already stated “Partitioning of rainfall”.
Response: Agree. It was removed.

Line 37: What do you mean by “hydrologic intensification”? Please, clarify.
Response: We rephrased it to be "due to hydrologic intensification, or acceleration of the water cycle".

Line 39: Not only by climate change, but also by restrictions imposed by water users associations. I would include “amongst other factors” at the end of this sentence.
Response: Good point. We added " amongst other factors" at the end of the sentence.

Line 42: “and their surface patterns”, this is unclear from my point of view. What do you mean by “patterns”? Do you mean infiltration? Do you mean runoff? Do you mean spatial or temporal patterns? I would use the word “pathways”, but I am not sure that this is what you meant. Please, clarify this point.
Response: Good point. We rephrased it to make it more clear: "Thus, a precise understanding of the amount of rainfall or sprinkler-irrigation returned to the atmosphere by interception or reaching cropland soils and their pathways at the surface and through the subsurface are arguably valuable to sustainable agriculture."

Methods: In this section, you could indicate the final number of papers that you used in the meta-analysis. For instance, at the end of the section, say something such as “This search produced 138 papers”.
Response: Good point. We added "This search resulted in 138 papers." at the end of this paragraph.

Line 85: “the Tabula software correctly digitized the datasets” instead of “the Tabula software did not incorrectly digitized the datasets”.

Response: Good point. We corrected it as suggested.

Line 86: What do you mean by “spot-checks”?

Response: We agree it is confusing, so we deleted it and it now reads "all data was able to be manually reviewed by at least one of the authors.”

Lines 88-89: This sentence (“Since canopy…”) seems to be incomplete. Please, check.
Response: Good point. We changed it to "Canopy precipitation partitioning research, in general, has no consensus on standard units for reporting throughfall, stemflow, and interception."

Line 91: Why mentioning Table S2 prior to Table S1?
Response: Good point. We re-numbered the table to reflect the logical order.

Line 103: “selected” instead of “select”.
Response: Good catch! Corrected.

Line 115: The Kruskal-Wallis test must have been mentioned in the previous section, when describing the analysis of data that you performed.
Response: We added the description of the test in line 103.

Figure 1: In the legend of the bottom right panel, use “Precipitation” instead of “Precipiattion”.
Response: Good catch! Figure 1 has been updated to reflect the change.

Line 143: I suggest swapping the order of the supplementary tables.
Response: Good point! Swapped the order as suggested.

Line 184: “reduce small rainstorms”, what do you mean? The intensity of the rainfall? The speed in which raindrops reach the soil surface? The volume of rainfall reaching the soil surface? Please, clarify.
Response: Good point. We changed it to "to intercept 92-94% of small rainstorms ".

Line 186: “stages” instead of “stage”.
Response: Corrected.

Line 189: Include “fractions” after “greater water”.
Response: Corrected.

Line 190: “crop growth” or “crop development” instead of “growth stage”.
Response: Corrected to “crop growth”.

Lines 221-223: Please, check this sentence. It seems to be incorrect. Maybe, remove “Observations of”.
Response: Good point. Deleted “Observations of”.

Line 230: “surface seals”, do you mean “surface crust”?
Response: We would prefer to use the term “seal” rather than “crust,” due to the differences in their definition by the UN-FAO (e.g., http://www.fao.org/3/T1696E/t1696e06.htm), where a seal is defined as “the orientation and packing of dispersed soil particles which have disintegrated from the soil aggregates due to the impact of rain drops.” (which is what we are discussing); however a crust is defined as “surface layer of the soil, ranging in thickness from a few millimetres to a few centimetres, which is much more compact than the material beneath. Structural crusts are formed also by physical forces as a result of trampling by livestock or through traffic by agricultural machinery and other vehicles.”

Line 266: I suggest removing “show clear differences in rainfall partitioning across vegetation types”.
Response: Good point. Changed it to read “show rainfall partitioning patterns across vegetation types.”

Line 280: If “theory” is the subject, then it should be “has been developed” instead of “have been developed”.

Response: Good catch. Changed it to “has been developed”.

Line 288: I suggest removing “(all that we were able to find)”.
Response: Removed as suggested.

Lines 317-318: This reference is incomplete. Volume and page numbers are missing.

Response: Revised as suggested.

Lines 322-324: This reference is incomplete. Volume and page numbers are missing.

Response: Revised as suggested.

Lines 335-337: This reference is incomplete. Page numbers are missing. If it is a book, publisher and location of the publisher are also missing.

Response: The publisher is indicated “C. Krebs” and we have added the location of the publisher.

Lines 338-339: This reference is incomplete. Page numbers are missing. If it is a book, publisher and location of the publisher are also missing.

Response: Revised as suggested.

Line 354: Page number is missing.
Response: Revised as suggested.

Line 357: “Theobroma cacao” should be in italics.
Response: Revised as suggested.

Lines 365-366: This reference is incomplete. Page numbers are missing. If it is a book, publisher and location of the publisher are also missing.

Response: Revised as suggested.

Line 391: “Coffea arabica” should be in italics.
Response: Revised as suggested.

Line 402: “Coffea arabica” should be in italics.
Response: Revised as suggested.

Lines 409-410: Please, do not use capital letters for the surnames of the authors. Include number of pages.

Response: Revised as suggested.

Line 431: “Punica granatum” should be in italics.
Response: Revised as suggested.

Line 484: “Theobroma cocoa” should be in italics.
Response: Revised as suggested.

Line 488: “Cocos nucifera” should be in italics.
Response: Revised as suggested.

Line 492-493: Please, check this reference. The abbreviated title of the journal and the number of pages seem to be incorrect.

Response: Revised as suggested. Please note that this journal no longer provides page numbers, just the article DOI.